# Immunopathogenesis of canine chronic ulcerative stomatitis

J. G. Anderson[1]*, A. Kol[2], P. Bizikova[3], B. P. Stapelton[4], K. Ford[4], A. Villarreal[2], R. J. Jimenez[2], D Vasilatis[2], B. G. Murphy[2]

1 Sacramento Veterinary Dental Services, Rancho Cordova, California, United States of America,
2 Department of Pathology, Microbiology and Immunology, School of Veterinary Medicine, University of California Davis, Davis, California, United States of America, 3 Department of Clinical Sciences, College of Veterinary Medicine, NC State University, Raleigh, North Carolina, United States of America, 4 Barrington Animal Hospital, Barrington, Illinois, United States of America

⊚ These authors contributed equally to this work.
* jgadvm@gmail.com

## Abstract

Canine Chronic Ulcerative Stomatitis is a spontaneously occurring inflammatory disease of the oral mucosa. An immune-mediated pathogenesis is suspected though not yet proven. We have recently reported on the clinical and histologic features, and identification of select leukocyte cell populations within the lesion. A clinical and histologic similarity to oral lichen planus of people was proposed. In the present study, these initial observations are extended by examining lesions from 24 dogs with clinical evidence of chronic ulcerative stomatitis. Because dogs with chronic ulcerative stomatitis often have concurrent periodontal disease, we wondered if dental plaque/biofilm may be a common instigator of inflammation in both lesions. We hypothesized that dogs with chronic ulcerative stomatitis would exhibit a spectrum of pathologic changes and phenotype of infiltrating leukocytes that would inform lesion pathogenesis and that these changes would differ from inflammatory phenotypes in periodontitis. Previously we identified chronic ulcerative stomatitis lesions to be rich in FoxP3+ and IL17+ cells. As such, we suspect that these leukocytes play an important role in lesion pathogenesis. The current study confirms the presence of moderate to large numbers of FoxP3+ T cells and IL17+ cells in all ulcerative stomatitis lesions using confocal immunofluorescence. Interestingly, the majority of IL17+ cells were determined to be non-T cells and IL17+ cell frequencies were negatively correlated with severity on the clinical scoring system. Three histologic subtypes of ulcerative stomatitis were determined; lichenoid, deep stomatitis and granulomatous. Periodontitis lesions, like stomatitis lesions, were B cell and plasma cell rich, but otherwise differed from the stomatitis lesions. Direct immunofluorescence results did not support an autoantibody-mediated autoimmune disease process. This investigation contributes to the body of literature regarding leukocyte involvement in canine idiopathic inflammatory disease pathogenesis.

## Introduction

Chronic ulcerative stomatitis (CCUS) is a chronic, progressive and painful oral inflammatory disease that is poorly responsive to current therapeutic approaches and has an incompletely

**Data Availability Statement:** All relevant data are within the manuscript and its Supporting Information files.

**Funding:** This study was supported by grant number The Waltham Foundation, TC 03102016,

The Waltham Foundation - https://www.waltham.com/grants-awards/, to JGA, BGM, AK. The funders had no role in study design, data collection and analysis, decision to publish, or preparation of the manuscript.

**Competing interests:** The authors have declared that no competing interests exist.

understood pathogenesis. We have previously described the clinical, radiographic, clinicopathologic and histopathologic characteristics of this disease.[1] We have further provided a preliminary and qualitative characterization of the inflammatory cell infiltrate that characterizes CCUS lesions. Specifically, we showed that CCUS lesions are rich in both B and T lymphocytes, as well as IL-17 producing cells. The current study is focused on further elucidating the immuno-pathogenesis of this poorly understood and devastating disease.

Further use of the pseudonym Canine Ulcerative Paradental Stomatitis has been discouraged as approximately 40% of the lesions occur adjacent to edentulous areas and are therefore not paradental. The clinical findings of CCUS are characterized by painful oral mucosal ulcers of varying size, pattern, appearance, and distribution. The disease is refractory to common medical therapies, and many clinicians have resorted to full-mouth tooth extraction as a salvage measure. We have introduced a clinical scoring system, called the Canine Ulcerative Stomatitis Disease Activity Index (CUSDAI) in an attempt to quantify disease severity and response to various therapeutic regimens "S1 Table".[1] CCUS histopathology typically features a densely arrayed inflammatory cell infiltrate at the interface between the superficial mucosal epithelium and subepithelial connective tissue (lichenoid infiltrate) that is composed of B cells, plasma cells, CD3 T cell subsets (including FoxP3+), neutrophils, and fewer mast cells within the deep submucosa.[1] An initial evaluation using multi-color immunofluorescence (IF) on a small number of CCUS cases, revealed moderate numbers of infiltrating IL17 + cells; interestingly the majority of these cells were determined to be CD3 negative.[1] Given the fact that CCUS patients respond poorly to current medical treatment strategies, there is a strong interest to better define the underlying pathogenesis of CCUS.

Detailed description of the various inflammatory cell infiltrates is a critical first step in understanding the pathogenesis. Since expansion of the T helper 1 (Th), Th2 paradigm, the relevance of Th17 cells, and their integral control of the tissue-based immune response alongside T regulatory (Treg) cells has been foundational.[2, 3] The IL17 family of cytokines are potent inducers of inflammation and help to protect against multiple microbial pathogens. However, these cytokines may also contribute to the tissue destruction that occurs in chronic inflammatory and immune-mediated diseases. The potent IL17 cytokine is secreted by specialized CD4 T cells, as well as other T cells, innate-like lymphocytes and other innate immunity cell types.

The balance between pro-inflammatory IL17 producing cells and immunoregulatory Treg cells in various types of inflammation and autoimmune diseases continues to receive intense investigation.[4, 5] Treg cells are thought to be critical regulators of the immune response and T-cell tolerance. It is postulated that in the human oral inflammatory disease, oral lichen planus (OLP), Treg cells are functionally deficient, and thereby fail to control the initiation and maintenance of this autoimmune disease.[6] In dogs, decreased numbers and function of Treg cells in inflammatory bowel disease and intestinal lymphoma has been shown to contribute to pathogenesis.[7, 8] In CCUS lesions, infiltrating FoxP3+ cells have been proposed to play a role in lesion pathogenesis.[1] Whether these lesion-associated Tregs are functionally deficient or play some kind of balancing role with proinflammatory IL17+ cells, awaits clarification.

In this CCUS study, tissues from dogs with periodontal disease (PD) were utilized as a comparator lesion as PD is one of the most common oral inflammatory diseases in dogs,[9] and often occurs concurrently in dogs with CCUS. Although the anatomic locations of CCUS lesions are known to vary, it was initially observed that dogs had lesions within the buccal mucosa adjacent to teeth (paradental). Many of these "paralesional" teeth had attached plaque biofilm or mineralized calculus, suggesting the possibility that the CCUS lesion might share pathologic features with the biofilm-associated PD lesion. In both humans and dogs, periodontitis is an important chronic inflammatory disease triggered by the oral microbiome and

altered by the function or dysfunction of the host's innate, and/or adaptive immune system. [10, 11] Given that the role of periodontal disease in the pathogenesis of CCUS is not fully understood, we have included a PD control group in an effort to shed light on potentially shared important pathogenic mechanisms.

Our goal for the present study was to better define the immunopathogenesis of CCUS by objectively quantifying inflammatory cell infiltrates, determining the pattern of immunoglobulins and complement fragment deposition within CCUS lesions and correlating these finding with clinical findings.

## Materials and methods

### Ethical statement

For all of the dogs in this study, tissue sampling was performed under general anesthesia (induction with Propofol 4 mg/kg and Valium 0.5 mg/kg intravenous, maintenance with isoflurane inhalant), with appropriate regional local anesthesia (Bupivacaine 0.5% ml amount dependent on animal size) and post-operative analgesic (Hydromorphone 0.5 mg/kg subcutaneous) administration. The study conformed to American Animal Hospital Association Guidelines for Dental Care. [12] Client consent forms were discussed and signed. Prospectively, the University of California, Davis IACUC counseled that since samples were obtained as part of dental treatment, an IACUC protocol was not required. No adverse events were documented as a result of tissue acquisition.

### Study design

This was a descriptive study of the leukocytic cell types within the mucosal tissues of client-owned dogs clinically diagnosed with chronic ulcerative stomatitis. Randomization was not applicable. Pathologists (BG Murphy and A Kol) were blinded to patient details and CUSDAI scores.

### Animals and clinical assessment

Twenty-four dogs with clinically evident CCUS were prospectively enrolled in this study from the clinical caseloads of the first (JG Anderson, 13 cases) and fourth (B Stapelton, 11 cases) authors "S2 Table". Patient guardians enrolled by both clinicians read and signed a patient release form "S3 Table". Animals were housed in their home environment. Inclusion criteria included two or more chronic erosions or ulcers in the buccal mucosal tissue opposite teeth or edentulous areas. Exclusion criteria included animals in renal failure, with previously diagnosed autoimmune or immune-mediated disease; and those dogs who were currently receiving immune-suppressing drug therapy. A CUSDAI score was determined and assigned and a generalized periodontal disease stage score (1 to 4) was determined for each case using clinical probing and full mouth dental radiographs.

For the CCUS patients, an approximately 10 X 10-mm section of affected oral mucosa was surgically removed while the patient was under general anesthesia. For some smaller lesions, this represented an excisional biopsy; and in other larger lesions, the biopsy was intralesional. The biopsy tissue was trisected, and sub-specimens were immersed in 10% buffered formalin, isotonic saline, or RNAlater Stabilization Solution (ThermoFisher Scientific). **Fig 1**. Tissues immersed in RNAlater were cooled on ice and frozen at -70$^{\circ}$ C within 24 hours after collection and were not further utilized in the present study. Tissues immersed in isotonic saline were cooled on ice and stored for less than 24 hours prior to embedding for frozen sections. Saline-immersed tissues were placed in OCT compound (Tissue-Plus, Fischer Healthcare) within a

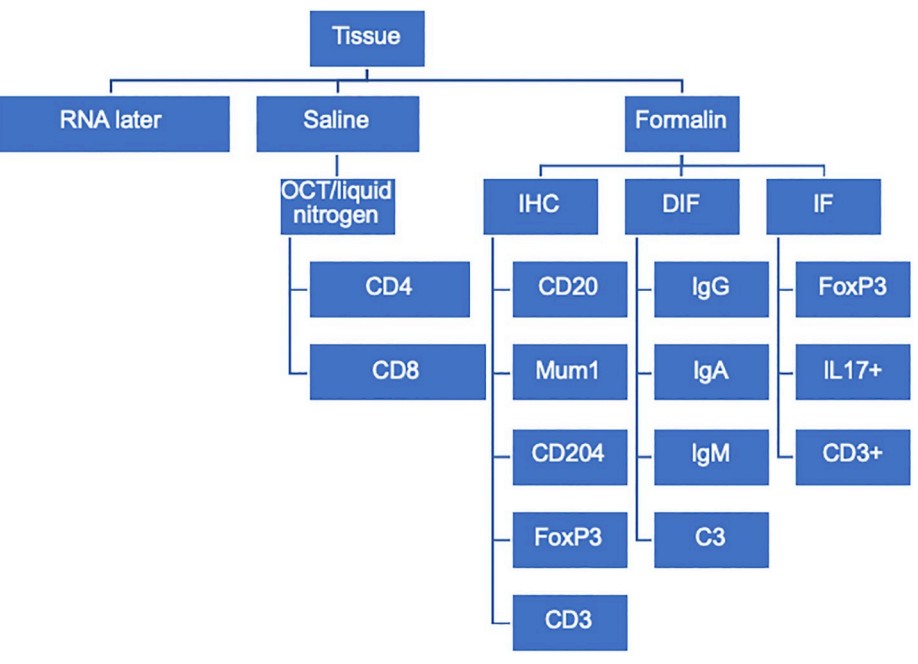

**Fig 1. CCUS sample distribution.**

plastic embedding cassette (Tissue-Tek Cryomold, Sakura) and flash frozen by suspending the cassette in 2-methylbutane (Fisher Chemical) cooled in a liquid nitrogen bath. The flash-frozen OCT embedded tissue was then stored at -70$^0$ C until sectioned on a cryomicrotome (Leica CM1860). The formalin treated mucosal tissues were fixed for a minimum of 48 hours, further bisected or trisected as needed, placed into plastic embedding cassettes and routinely processed for 4 micron-thick sections and affixed to positively charged glass slides (formalin-fixed paraffin embedded (FFPE) sections).

For the ten control periodontal disease tissues; one or more affected teeth and adjacent soft tissues (periodontal ligament and adjacent fragments of alveolar bone) were harvested from extracted teeth-bone specimens of dogs with clinical evidence of Stage 3 or 4 periodontitis. These samples were collected from the clinical caseload of the first author. After extraction, teeth and associated soft tissues were immediately immersed in 10% buffered formalin for a minimum of 48 hours and subsequently decalcified in 15% formic acid as needed to allow sectioning (48–96 hours). Decalcified tissues were processed and affixed to glass slides as described above. Normal canine lymph node tissue (n = 1) and canine buccal mucosa (n = 5) were obtained from dogs euthanized at the University of California Davis, Veterinary Medical Teaching Hospital for reasons unrelated to oral disease undergoing unrestricted necropsies. These tissues were fixed in 10% buffered formalin for a minimum of 48 hours, bisected and routinely processed onto positively charged glass slides as described above.

## Histopathology

Mucosal CCUS lesions, teeth/periodontal tissue, normal lymph node tissue and normal buccal mucosa tissue were stained with hematoxylin and eosin (HE) or toluidine blue according to standard protocols. Tissue sections were examined using brightfield microscopy and digital photographs were obtained using an Olympus BX41 microscope and Tucsen 16 MP digital camera (Tucsen Photonics).

## Immunohistochemistry

Immunohistochemistry (IHC) assays were performed on 4-micron thick, formalin-fixed, paraffin-embedded (FFPE) tissue sections mounted on positively charged slides and air dried overnight. Sections were immune-labeled for CD3 (T cells), CD20 (B cells), Mum-1 (plasma cells), CD204 (macrophage) or FoxP3 (Treg) cells. Specific antibodies and supporting citations are summarized in **Table 1**.

Sections were deparaffinized in xylene and rehydrated through graded alcohols to phosphate buffered saline (PBS). Endogenous peroxidases were quenched with 0.3% hydrogen peroxide for 30 minutes in methanol prior to rehydration. CD3, CD20 and FoxP3 slides underwent heat induced epitope retrieval (HIER) in Dako Target Retrieval Solution (Dako S1699) for 30 minutes at $95^o$ C and then cooled for 20 minutes. Mum-1 slides underwent HIER treatment in EDTA buffer (Thermoscientific AP-9004-500) for 30 minutes at $95^o$ C and then cooled for 20 minutes. CD204 slides underwent HIER treatment in EDTA buffer pH 9.0 (VMTH) for 30 minutes at $95^o$ C and then cooled for 20 minutes.

All of the tissue sections were blocked for 20 minutes in 10% normal horse serum diluted in PBS. Primary antibodies and dilutions were as follows: rat anti-CD3, diluted 1:10; rabbit anti-CD20 diluted 1:300; rat anti-FoxP3, diluted 1:100; rabbit anti-Mum1, diluted 1:100; mouse anti-CD204, diluted 1:200.

BioCare Medical mouse-on canine HRP Polymer (MC541L) was applied for 30 minutes to label mouse anti-CD204. BioCare Medical rabbit-on-canine HRP-Polymer (RC542L) was applied for 30 minutes to label rabbit anti-Mum1, and anti-CD20. BioCare Medical rat HRP-Polymer, 1-step mouse absorbed (BRR4016L) was applied for 30 minutes to label rat anti-CD3 and anti-FoxP3. All bound antibodies were visualized with NovaRed for peroxidase (Vector SK-4800), per manufacturer's instructions. Sections were counterstained in Hematoxylin 1 (Fisher HealthCare, 22-220-101) for 10 seconds and Bluing Reagent (Fisher HealthCare 220–106) for 2 minutes and air dried prior to attaching cover slips. A normal canine lymph node was used as the positive control tissue for the leukocyte phenotyping.

Immunohistochemistry to identify CD4 or CD8 T cells was performed on non-formalin fixed, flash-frozen tissue sections mounted on positively charged slides and air-dried overnight. Slides were quenched for 10 minutes in 0.1M phosphate buffered saline (PBS), pH 7.4 (PBS) with 0.3% hydrogen peroxide and 0.1% sodium azide. Slides were rinsed in PBS, then treated with 10% normal horse serum in PBS for 20 minutes. Without rinsing, the CD4 or CD8 alpha antibody diluted 1:10 in PBS was applied and incubated for 1 hour. As both antibodies are made in mouse, multiplexing is not feasible. Slides were rinsed with PBS, and

**Table 1. Primary antibodies used in this study.**

| Target antigen | Target species | monoclonal/polyclonal | Make | Catalogue number | Validation studies in dogs |
|---|---|---|---|---|---|
| CD3 | Human | Monoclonal (CD3-12) | Bio-Rad | MAC1477 | [13] |
| CD3 | Human | Polyclonal | Dako | GA50361 | [14, 15] |
| **CD4** | Canine | Monoclonal (1E4) | UC Davis Leukocyte Antigen Biology Lab, Dr. Peter Moore | N/A | [16] |
| **CD8** | Canine | Monoclonal (JD3) | UC Davis Leukocyte Antigen Biology Lab, Dr. Peter Moore | N/A | [16] |
| **FoxP3** | Mouse | Monoclonal (FJK-16s) | Invitrogen | 14-5773-82 | [17, 18] |
| **IL17** | Human | Polyclonal | R&D systems | AF-317 | [19] |
| **CD20** | Human | Polyclonal | ThermoFisher Scientific | RB-9013 | [20] |
| **Mum-1** | Human | Monoclonal (BC5) | BioCare Medical | CRM352B | [21, 22] |
| **CD204** | Human | Monoclonal (SRA-E5) | TransGenic | KT022 | [23, 24] |

biotinylated anti-mouse antibody diluted 1:500 in PBS was applied for 20 minutes. Slides were rinsed with PBS, and streptavidin-horseradish peroxidase diluted 1:500 in PBS was applied for 20 minutes. Bound antibodies were visualized with Vector's NovaRed, per manufacturer's instructions. All incubations occurred at room temperature. Sections were counterstained with Hematoxylin 1 (Fisher HealthCare,22-220-101) for 2 minutes, followed by Bluing Reagent (Fisher HealthCare, 220–106) for 2 minutes, air dried followed by coverslip application.

All of the IHC assays were examined by a single pathologist (BGM). The semiquantitative scoring system was based upon published standard. [25] The semi-quantitative scoring systems for canine mucosal mast cells, FoxP3+ and CD3+ T cells have been previously published. [1] Semiquantitative ordinal scoring systems were developed for Mum1, CD4, CD8, CD20 and CD204 and were based upon the percent of the surface area of a representative high power (400x) magnified field that stained positively (IHC scoring system **Table 2**). A normal canine mandibular lymph node and normal oral buccal mucosa were used as positive control tissues for these leukocyte IHC stains. For each submitted tissue sample, the pathologist was blinded to the signalment and clinical severity score.

## Immunofluorescence and direct immunofluorescence

Samples were stained for immunofluorescence (IF) as previously described. Briefly, FFPE tissue sections were routinely deparaffinized in xylene and serial ethanol dilutions, following heat-induced antigen retrieval (antigen retrieval buffer; Dako). Samples were further blocked with normal donkey serum and FcR Blocking Reagent (Miltenyi) and incubated overnight with rabbit anti-human CD3 polyclonal antibody (1:50 dilutio) and goat antihuman IL17 polyclonal antibody (1:100 dilution) or rat anti-FoxP3 (1:200 dilution). Twenty-four hours later, slides were extensively washed and treated with donkey anti-rabbit Alexa Fluor 488 and

**Table 2. IHC Semi-quantitative scoring system.**

| [1]Mast cells (T. blue)- # of cells/200x field | Scoring System |
|---|---|
| none | 0 |
| 1-4/200x field | 1+ |
| 5-20/200x field | 2+ |
| 20+/200x field | 3+ |
| | |
| [1]FoxP3 (IHC)- # cells/200x field | |
| none | 0 |
| 1-5/200x field | 1+ |
| 6-20/200x field | 2+ |
| 21+/200x field | 3+ |
| | |
| [1]T cells (CD3 IHC)- % surface area (400x)<br>CD4 (frozen IHC)<br>CD8 (frozen IHC)<br>[1]B cells (CD20 IHC)<br>Plasma cells (mum1)<br>Macrophages (CD204) | |
| none | 0 |
| <20% of 400x field | 1+ |
| 20–50% of 400x field | 2+ |
| 51% + of 400x field | 3+ |
| | |

donkey anti-goat Alexa Fluor 555 or donkey anti-rat Alexa Fluor 594 (ThermoFischer Scientific). All secondary antibodies were used at 1:400 dilution. Finally, nuclei were labeled with DAPI (ThermoFischer Scientific). All immunohistochemistry and immunofluorescence studies had several layers of negative and positive controls to ensure appropriate interpretation of our findings. Specifically, we have used a 'No antibody' control to determine the level of tissue auto-fluorescence and 'No primary antibody control' to determine the level of non-specific secondary antibody binding. For positive controls we have used canine lymph node sections which contain the various immune cell subsets. Moreover, cell subset distribution within the lymph node tissue was further used to evaluate our positive controls. Five random images magnified at 200x were acquired from each section with a super-resolution confocal microscope (Leica) and analyzed using image analysis software (Image J, NIH and lmaris, Oxford Instruments). A "reactive" (hyperplastic) canine mesenteric lymph node was used as a positive control. The negative control slides omitted the primary antibodies.

For direct immunofluorescence (DIF), formalin-fixed, paraffin-embedded biopsy sections of buccal mucosa from 17 dogs diagnosed with CCUS were available for detection of tissue-bound IgG, IgM and IgA antibodies and complement C3 using DIF. Sections of buccal mucosa from five healthy dogs served as controls. All tested tissues were deparaffinized, rehydrated and digested with 0.1% trypsin (MilliporeSigma, St. Lois, MO, USA) for 30 minutes at 37 Celsius. Tissues were rinsed in PBS, subsequently blocked with 1% newborn calf serum (MilliporeSigma, St. Lois, MO, USA) and the sections were incubated at room temperature for 30 minutes with either rabbit anti-canine IgG (1:400; Alexa Fluor-488 conjugated, Jackson ImmunoResearch, West Grove, PA, USA), goat anti-canine IgM (1:200; FITC conjugated, Bethyl, Montgomery, TX, USA), goat anti-canine IgA (1:400; FITC conjugated, Bethyl, Montgomery, TX, USA) or goat anti-canine C3 (1:100; FITC conjugated, Bethyl, Montgomery, TX, USA) antibodies. Lymph node and mucosal sections from CCUS and healthy dogs in which antibody was replaced by PBS served as positive and negative controls, respectively. Finally, all tissues were mounted with Vectashield Antifade mounting medium with a nuclear counterstain DAPI (Vector Laboratories, Burlingame, CA, USA). Sections were evaluated using a fluorescent microscope and representative images were obtained. The oral mucosa tissues were evaluated for presence of immunostaining within the mucosal epithelium (intercellular, intracellular or nuclear), at the basement membrane level (continuous, patchy) and within the lamina propria. The staining intensity was graded subjectively as 0 = none, 1 = mild, 2 = moderate, 3 = strong.

## Statistical analyses

All statistics were performed with GraphPad Prism 7.0. Data was tested for normality with the Shapiro-Wilk test to inform the use of parametric vs non-parametric methods in downstream analysis. A non-parametric Kruskal-Wallis ANOVA test was used to identify differences in within a group of three or more means. Where differences were identified, non-parametric t tests (Mann-Whitney test) were used to explore differences between means. Spearman's rank correlation co-efficiency was used to determine association between sex, CUSDAI scores, disease variants and leukocyte indices. A P value < 0.05 was considered to be significant.

## Results

### Histopathology

The morphologic diagnoses for all of the CCUS lesions were consistent with a submucosal interface mucositis (pleocellular stomatitis) as previously described. (1) However, three relatively distinct histologic subtypes of CCUS emerged, i) granulomatous inflammation featuring a predominance of histiocytic cells (n = 3), ii) lichenoid inflammation (interface mucositis)

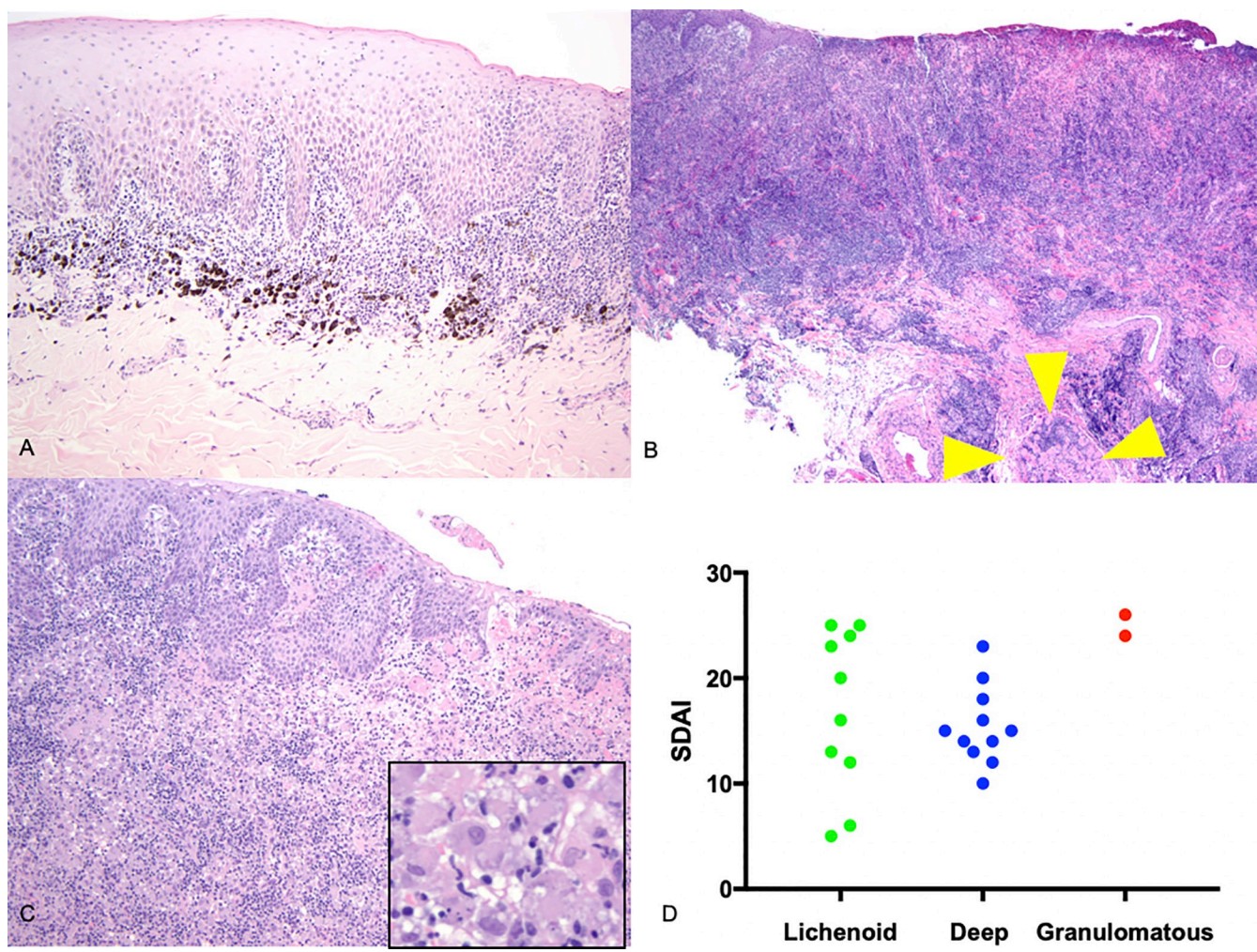

**Fig 2. Histologic subtypes of CCUS.** A-Lichenoid stomatitis-CASE G 100x; B-Deep stomatitis-CASE V 40x (yellow 445 arrowheads-inflamed skeletal muscle); C-Granulomatous stomatitis-CASE U 100x (inset-446 histiocytic inflammation, 200x) D-Clinical score (CUSDAI) versus CCUS histologic subtype.

(n = 12), and iii) deep stomatitis in which the inflammatory infiltrate extended down into the deep buccal mucosa and skeletal muscle of the lip (n = 10). (**Fig 2A–2D**) In all CCUS subtypes, the overlying epithelium was either eroded, ulcerated and/or hyperplastic and lymphoid follicles were infrequently present in the submucosal connective tissue. The pleocellular lesions generally featured an abundance of infiltrating neutrophils, lymphocytes and plasma cells forming sheets and clusters. Small numbers of mast cells were typically identified in the deep submucosa while eosinophils were not identified. The number of infiltrating macrophages in the CCUS lesion varied by subtype and macrophages containing black melanin granules were infrequently present within the superficial lamina propria. The three histologic subsets did not differ in their mean clinical index of severity (i.e.CUSDAI), though the only two granulomatous lesions came from dogs with high CUSDAI scores (24 and 26).

## Immune cell infiltrate

Subjectively, CCUS lesions appeared to have a predominance of infiltrating B cells and plasma cells over T cells and T cell subsets. However, the IHC-based semi-quantitative scoring system

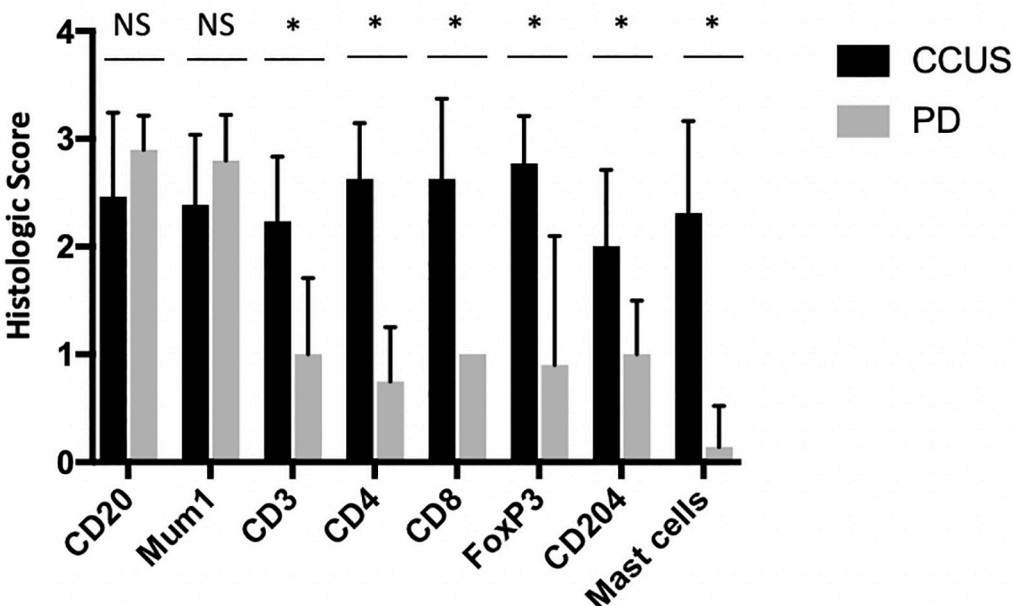

**Fig 3. Leukocyte subsets- CCUS versus periodontitis controls.**

indicated that the ordinal numbers of all of these cell types were statistically inseparable (Fig 3). There were no significant differences between the mean IHC scores for any of these leukocyte subsets within the CCUS lesions.

B cells and plasma cells were present in the lamina propria and in lymphoid follicles (when present) but did not infiltrate into the epithelium. CD3+ T cells and FoxP3+ were most numerous in the lamina propria and occasionally infiltrated into the overlying mucosal epithelium. CD204+ cells (macrophages) were less well represented in the CCUS lesion with a mean ordinal score of 1.7. Macrophages were generally present at the interface of the deep lamina propria and the muscular layer of the lip. However, in the granulomatous variant of CCUS, macrophages were numerous (semiquantitative score of 3), tended to form expansive sheets and were superficially located. Neutrophils were abundant in most of the CCUS lesions, especially subjacent to ulcerated regions. Mast cells were moderately common in the CCUS lesion with an average of 5–20 cells/per 200x field and were generally located deep within the lamina propria. Within the CCUS population, there were 14 cases with Stage 1 or 2 PD, two with Stage 3 PD, and ten with Stage 4 PD. Differences in the numbers of infiltrating mast cells and CD204+ cells, between these groups were not statistically significant.

All immune cell subsets that were investigated (i.e. B lymphocytes, T lymphocytes, plasma cells, T helpers, cytotoxic T cells, macrophages and Treg) were increased in CCUS lesions compared with healthy controls. While infiltrating B cells and plasma cells did not differ between PD lesions and CCUS lesions, the rest of the tested immune cells were higher in CCUS.

## Comparative results

In the normal buccal mucosa, small numbers of mast cells, CD3, CD4, CD8 and CD204+ cells were identified within the lamina propria while very few B cells (CD20 or plasma cells) were identified and FoxP3 + cells were not identified. The number of infiltrating leukocytes was significantly higher in the CCUS lesions relative to the normal canine buccal mucosa. The complete data set is available, "**S4 Table**".

In the tooth-associated periodontitis lesions, the inflamed periodontal tissue often represented a small proportion of the histologic section, and there were difficulties with the IHC stained sections "washing off the slides". Therefore, only ten of fifteen PD cases were examined with the slate of immunohistochemical assays. In these ten examined PD lesions, the minimal periodontal tissues attached to the tooth root cementum was multifocally infiltrated by large numbers of lymphocytes and plasma cells and fewer neutrophils. Interestingly, very few CD3, CD4, CD8, CD204, FoxP3+ and mast cells were identified in the periodontitis lesions. In comparison to the CCUS lesions, CD3, CD4, CD8, mast cells, CD204 and FoxP3+ cell numbers in periodontitis lesions were significantly less, while the numbers of infiltrating CD20 and Mum1+ plasma cells were indistinguishable between these two lesions.

## Immunofluorescence

The 23 examined CCUS lesions had markedly increased numbers of CD3+ T cells within the epithelial-lamina propria interface relative to normal canine oral mucosa. (**Fig 4**) IL17+ cells were not identified in the normal control mucosa. CD3+ T cells expanded both the lamina propria and infiltrated into the epithelium, with a median of 158 CD3+ T cells per 20x field (range 6–731). As expected, FoxP3 staining was localized to nuclei of cells that were also CD3+ (i.e. T cells). FoxP3+ T cells were noted in all but one of the samples (n = 16) and have accounted for 4.1% of the T cells in the sample (range 0.0–17). Strong cytoplasmic expression

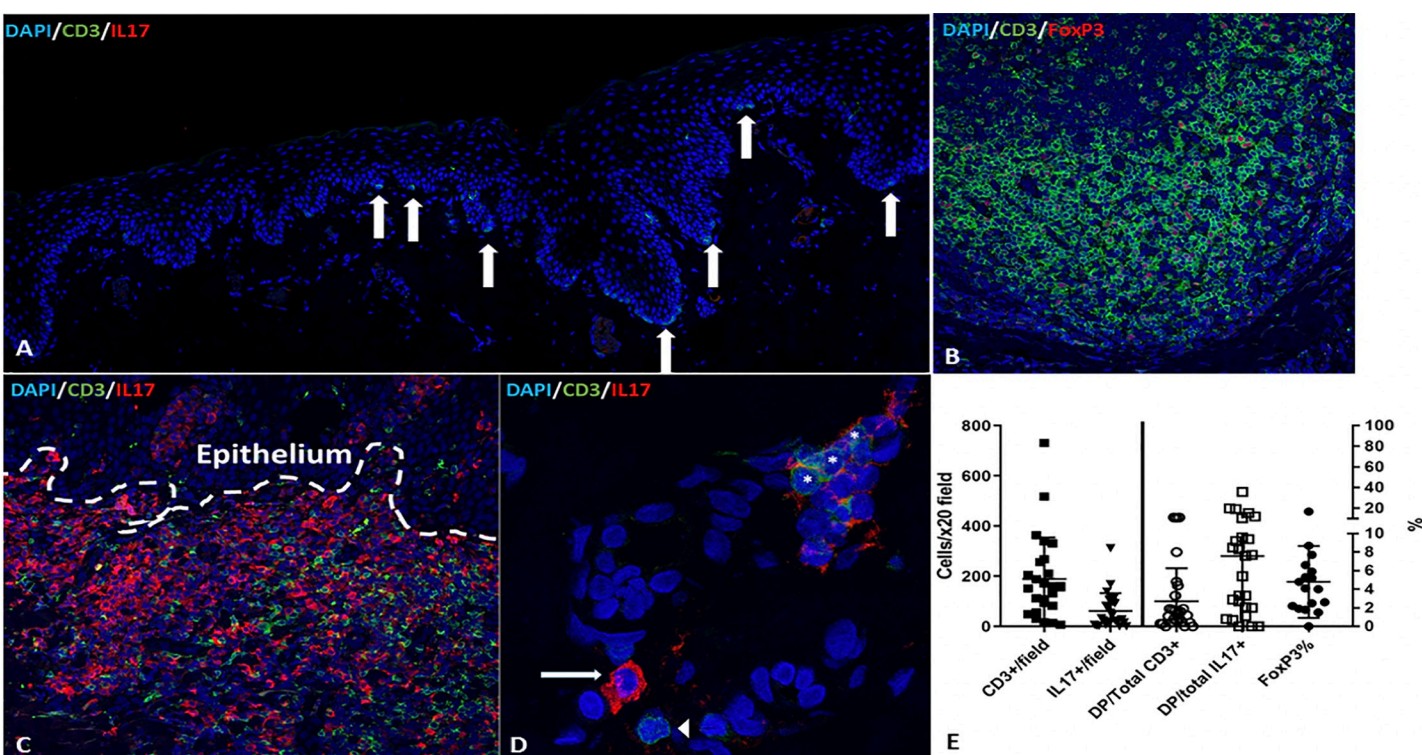

**Fig 4. T helper subset distribution in CCUS lesions.** A -Normal canine oral mucosa illustrates very few CD3+ T cells (white arrows) that are distributed at the mucosal–submucosal interface (A, stitched 20x tile image); B -Co-labeling of CD3 (green) and FoxP3 (red) reveals abundant FoxP3+ T cells that are noted with the submucosa of CCUS lesions; C-IL17 (red) and CD3(green) co-staining reveals marked infiltration of sub-epithelial and mucosal tissues with CD3+T cells and IL17 producing cells. Note that most of the IL17 producing cells (i.e. red) are not double labeled with CD3 (i.e. green) indicating that IL17 producing cells are primarily non-T cells. D—Close up (630x) image illustrates a single positive, IL17, cell (white arrow), a single positive CD3 T cell (white arrow head) and a group of double positive (i.e. IL17/CD3) IL17 producing T cells (asterisk). E—No significant differences were identified in the frequencies of CD3+/FoxP3+ cells, CD3+ cells, IL17+ cells and DP positive cells between the three histologic variants.

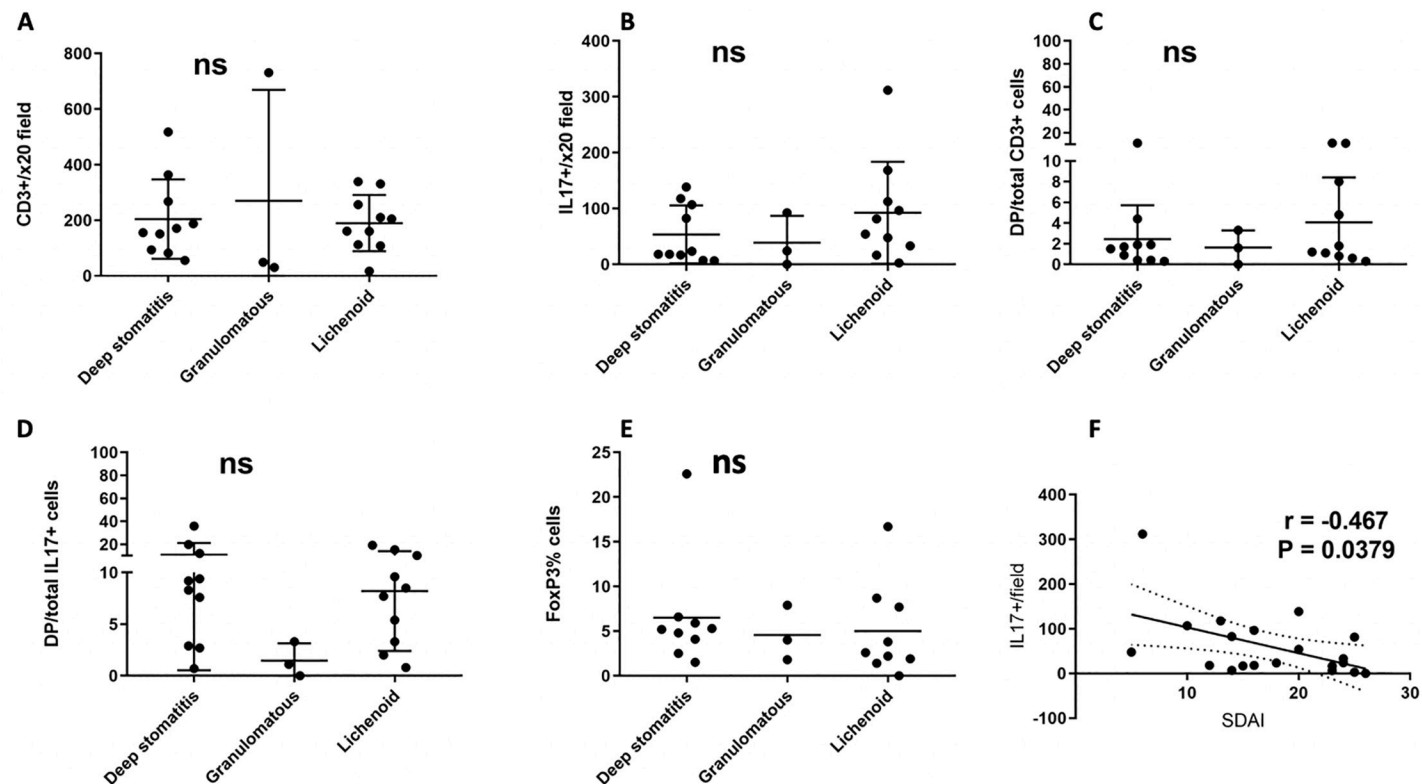

**Fig 5. T cell and IL17 producing cell subsets in CCUS histologic types.** A-There was no statistically significant difference in the number of infiltrating T cells; B-IL17 + cells; C and D-percent double positive (IL17+/CD3+); and E-Treg cells. F-A negative correlation was found between the numbers of IL17+ cells per 20x field and the stomatitis disease activity index.

of IL17 in scattered IL17+ cells was noted in all but one of the samples (n = 22) with a median of 47.8 cells/20x field (range 0–312). Consistent with previous findings, we noted that very few cells were CD3+/IL17+ double positive (DP). DP cells accounted for a median of 1.6% of the CD3+ T cells (range 0–11) and a median of 7.7% of the IL17+ cells (range 0–36). (**Fig 4**) While the CD3+/IL17+ cells are likely to represent activated Th17 cells, the lineage of the CD3-/IL17 + cells was not determined. Many cells with segmented nuclei (granulocytes) were present and were determined to be IL17 negative.

There was no significant difference between the three histologic CCUS subsets in the number of infiltrating T cells, IL17+ cells, the proportion of IL17+/CD3+ out of the total CD3+, the proportion of IL17+/CD3+ out of the total IL17+ and number of infiltrating Treg (**Fig 5**). A significant negative correlation (r = -0.467, p = 0.038) between the frequency of IL17+ cells in the lesions and the CUSDAI was identified, suggesting that IL17 may have a physiologic/protective role in this disease process.

## Direct immunofluorescence

Unique DIF staining patterns associated with known auto-immune skin diseases such as deeppemphigus, autoimmune subepidermal blistering skin diseases or lupus erythematosus were not detected in CCUS lesions. Positive intracellular IF staining for different antibody classes was seen within the lamina propria where it was associated with the infiltrating mononuclear cells. IgG deposits were most numerous, followed by IgA and IgM. There was minimal C3 deposition.

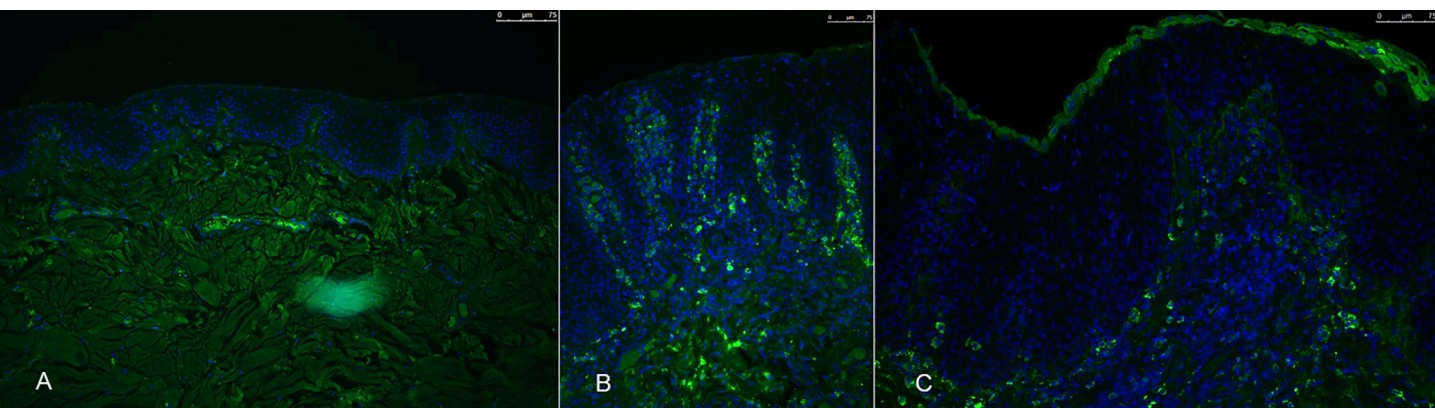

**Fig 6. A-C** Direct Immunofluorescence in health and CCUS, IgG. A-DIF–tissue-bound IgG–using gingival mucosa from a healthy dog (A) or dog with CUS (B, C); B-A band of IgG positive cells in the lamina propria below the mucosal epithelium (green) with scattered IgG positive cells in deeper layers of the lamina propria; C-In rare dogs, in addition to the IgG positive mononuclear cells in the lamina propria, a faint and patchy basement membrane zone deposit could be appreciated. Blue nuclear staining = DAPI.

Patchy to diffuse IgG staining was seen in the epithelial mucosa of 7/17 cases, and 3+ IgG staining was present in all cases. One case revealed patchy IgG staining in the basement membrane zone (BMZ). There was no IgA found within the epithelial mucosa, and from rare to 3 + staining in the submucosa of 13/17. Rare IgM was found in the epithelial mucosa of 1 case, in the submucosa, no specific staining (NSS) was seen in 3 cases, rare in seven cases, 1+ in six, 2+ in 2 and 3+ in 1. Complement staining in the mucosa was seen in 1 case at 1+, rare in the submucosa of 6, 1+ in two cases, 2+ in one case, and NSS in 9. (Fig 6A–6C), (Fig 7A–7D)

## Discussion

CCUS is a pleocellular chronic inflammatory mucosal disease that includes three distinct histologic subtypes. The histologic subtypes of CCUS, lichenoid stomatitis, deep stomatitis and granulomatous stomatitis, may either represent a continuum disease process or three distinct disease variants. The mean clinical disease score was most severe for the granulomatous variant (25/32). CCUS lesions feature a wide variety of infiltrating leukocytes with moderate to large numbers of T cells (CD3, CD4, CD8, FoxP3), B cells (CD20, Mum1), IL17+ cells, macrophages (CD204) and mast cells. Though lesion associated neutrophils were not quantitated, they were present in moderate to large numbers in most of the examined CCUS lesions.

Both FoxP3+ and CD3-/IL17+ cells are involved in CCUS pathogenesis. However, there were no significant differences in the relative numbers of IL17+ cells and CD3+/IL17+ double positive cells, nor in the cells of the different histologic subtypes, further obscuring the role of IL17. The relatively high levels of Tregs (i.e. FoxP3+ T cells) in CCUS, did not vary between patient clinical severity scores, clinical periodontitis scores, histologic subtypes, gender, or CD3-/IL17+ cells. Whether Tregs in CCUS lesions play an anti-inflammatory role, alter immunosuppression or disrupt mucosal tolerance remains to be determined.

Th17 cells have emerged as crucial regulators of tissue homeostasis and immunopathology at the oral mucosal barrier.[26] The IL17 family of cytokines are known to be potent inducers of inflammation and to help protect against invading pathogens; however, they can promote autoimmune disease when working together with Th1 cells and accompanied by chronic immune and inflammatory processes. At this point in our understanding of CCUS pathogenesis, the lesion appears to feature infiltration of mostly CD3 negative IL17+ cells. Neutrophils, though numerous in the interface mucositis, seem to be IL17 negative. Although multiple

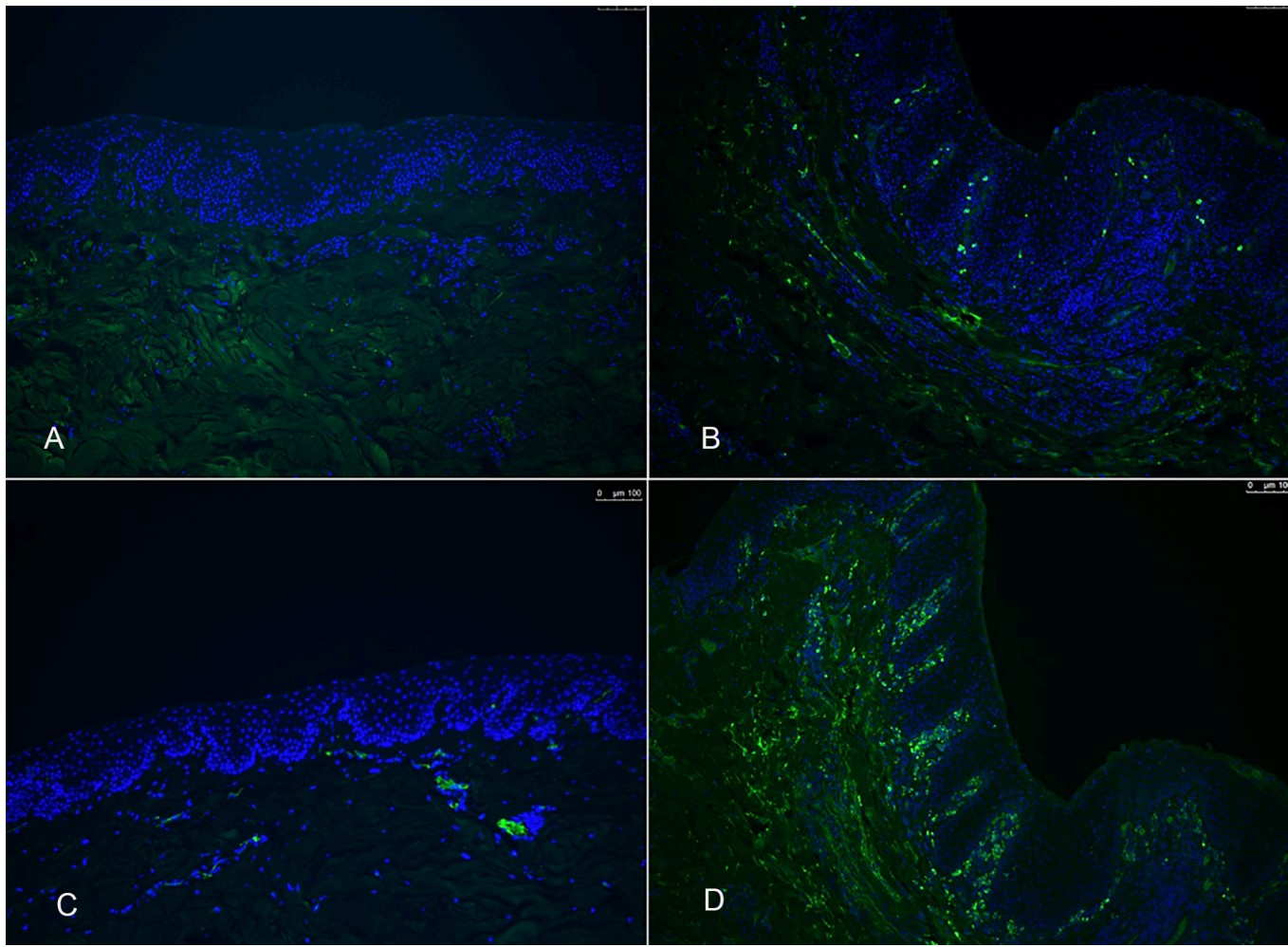

**Fig 7. A-D** Direct Immunofluorescence in health and CCUS, IgM and IgA. DIF–tissue-bound IgM (A-B) and IgA(C-D)–using gingival mucosa from a healthy dog (A, C) or dog with CUS (B, D). Positively stained mononuclear cells in the lamina propria and within mucosal epithelium depicted in green. Blue nuclear staining = DAPI.

studies indicate that IL17 is primarily produced by CD4+ Th cells;[26] numerous other cell types appear to be capable of producing IL17 including γδ T cells,[27, 28] natural killer T cells (NKT), neutrophils,[29] monocytes, mast cells, dendritic cells, vascular smooth muscle cells [26] and by follicular helper CD4 T ($T_{fh}$) cells.[30] Confocal microscopy indicates that IL17 + cells in CCUS lesions are mononuclear in origin; thereby eliminating granulocytes. However, the precise determination of the IL17+ cell phenotype in CCUS lesions is a critical future goal for a more complete understanding of disease pathogenesis.

Current CCUS data suggests that IL17+ cells may, in part, play some kind of protective role as the clinical severity score is inversely correlated to the number of IL17+ cells within the lesion. That is, as the clinical severity score decreased, the number of lesion-associated IL17 + cells increased.

The role of IL17 in periodontal disease in people is uncertain, since IL17 has been shown to promote bone destruction but is nonetheless essential to protect the host from pathogens, including periodontopathic organisms.[31] Defects in Th17 cells and/or their signature cytokines result in significant susceptibility to oral fungal infections.[32–34] As well, amplified/

uncontrolled Th17 responses have been documented in periodontitis.[35–37] Recent evidence has shown that Th17 cells are more osteoclastogenic than other T helper subsets such as Th1 or Th2. There is also evidence that the periodontopathogen *Porphyromonas gingivalis* promotes an IL 17 environment in order to obtain nutrients from tissue breakdown products.[35, 38, 39] IL17 sits at the center of many complex diseases that integrate innate and adaptive immune mechanisms and requires careful study to maximize host protective effects and minimize host deleterious effects.

PD is the most common chronic inflammatory disease of dogs and often occurs in conjunction with CCUS lesions. In contrast to the highly pleocellular CCUS lesions, the independent canine periodontitis lesions analyzed in this study were predominately characterized by a B cell infiltrate (CD20 and Mum1) but not by infiltrating T cells, macrophages, mast cells or FoxP3+ cells. PD is common in human patients and approximately 10% of the population will develop severe periodontal disease.[40, 41] Studies in people have demonstrated that development of periodontitis involves a switch from a gingivitis lesion, mainly mediated by T cells, to one predominated by B cells and plasma cells.[42] In PD lesions, B cells have been shown to have both protective and detrimental roles in settings of immunopathology.[43] The pathogenesis of human PD seems to involve an interplay between the tooth-associated bacterial biofilm and the host immune response. Severe periodontal destruction is associated with systemic translocation of periodontal microbes and is linked to numerous systemic inflammatory conditions, indicating that, in people, local immune/microbiome imbalance may affect systemic inflammatory processes, either through increased microbial translocation, systemic inflammation, or shared immunological mechanisms.[35] The composition of the plaque-associated microbiome in canines with ulcerative stomatitis has recently been determined, and similar conclusions may be relevant. We, as yet, do not know how many IL17+ cells are present within the non-CCUS periodontal disease lesions; though there seemed to be no difference in IL17 numbers between periodontal disease stages within the CCUS population. Implicit in our understanding will be to determine the source of the IL17 in CCUS; and for severe periodontitis if IL17 is present.

CCUS has multiple pathologic similarities to OLP in people.[1] OLP studies suggest FoxP3+Tregs have a more prominent role in lesion pathogenesis when compared to IL17+ cells.[44] The majority of FoxP3+ cells in OLP were identified in the sub-epithelial infiltrate, while IL17+ cells were found deeper in the stromal tissues.[45] As well, in human patients with clinically erosive OLP lesions, Foxp3 mRNA expression was significantly lower in circulating CD4+CD25+ T cells and tissue explants compared to patients with reticular lesions, and lowest in patients with a history of OLP of more than one year or with a history of relapse.[46] This study indicated that Foxp3 expression in patients with OLP was associated with the severity and duration of the disorder, suggesting altered immunosuppression in the development, clinical course and responsiveness to treatment. In another study, impaired suppressive function of CD4+ CD25+ T cells was demonstrated in OLP patients indicating that Tregs in OLP are frequently expanded but functionally deficient. The authors conclude that this may explain why the increased Tregs in OLP fail to control the pathogenesis and development of this autoimmune disease.[6] We propose for further investigation in CCUS that the high numbers of FoxP3 may as well be deficient.

Consideration of the pathogenic mechanisms operating in OLP may be useful to CCUS disease inquiry. At the cellular level, OLP may result from an immunologically induced apoptosis of the basal keratinocytes, due to cytotoxic CD8+ cell response on modified keratinocyte surface antigen. IF in OLP revealed that FoxP3+ cells co-localized with T cells. Double labelling immunofluorescence indicated co-localization of IL17 with tryptase (+) mast cells, solidifying their role in pathogenesis. As there are large numbers of FoxP3+ cells in CCUS as well as IL17

+ cells, we do not yet know which cell type is most important. Co-localization studies are planned to determine if IL17+ cells are mast cells, thereby suggesting a role for CCUS as an animal model for OLP.

One of the hallmarks of OLP diagnosis in human patients is "shaggy fibrinogen staining" at the basement membrane zone on direct immunofluorescence.[47] Unfortunately, fibrinogen DIF staining was not assessed in these CCUS lesions. DIF staining of the CCUS lesions did reveal different classes of antibodies, IgG, IgA, and IgM, and very little complement, found within mononuclear cells, which likely represent plasma cells infiltrating the lamina propria. None of the tissue specific staining patterns associated with known autoimmune skin diseases such as pemphigus, autoimmune subepidermal blistering skin diseases or lupus erythematosus were observed. This pattern supports the inflammatory nature of CCUS, and perhaps the result of directly cytotoxic T lymphocytes and not a classic auto-antibody immune mediated pathogenesis.

For future analyses, serum samples from CCUS cases were banked, as were lesion tissue samples for the determination of RNA expression. Flow cytometry or IF could be used to further identify the IL17-expressing cell phenotype. If the source for IL17 can be determined, then potential targeted therapeutics may be developed. Further clarification of the role of IL17 in this spontaneously occurring disease may provide a non-mouse model for other IL17 rich diseases. Determination of the oral microbiome from plaque samples of CCUS lesions is complete and analysis may reveal a dysbiotic trigger in mucosal immunopathology. Other studies are required to determine the function of FoxP3 in CCUS; and whether its exact role is physiologic or pathologic.

Limitations in the current study include that IL17 IF analysis was not evaluated in periodontitis lesions. Comparisons are therefore limited to the analyses performed.

## Conclusions

Three histologic subtypes of CCUS include granulomatous, lichenoid inflammation and deep stomatitis. Leukocyte numbers and distribution are consistent across the CCUS cases and differ from normal and periodontitis controls. CCUS is a heterogeneous, pleocelluar chronic inflammatory mucosal disease, and regulatory FoxP3+ and CD3-/IL17+ cells are involved in the pathogenesis. IL17 positive cells in CCUS are mononuclear and primarily CD3 negative.

IL17+ cell frequencies were negatively correlated with the clinical scoring system. Direct immunofluorescence did not support an autoantibody auto-immune disease process.

## Supporting information

**S1 Table. Canine Ulcerative Stomatitis Disease Activity Index.**
(DOCX)

**S2 Table. Case designation, patient sex, CUSDAI score, PD score, and institution.**
(DOCX)

**S3 Table. CLINICAL INVESTIGATION CLIENT CONSENT FORM: Immunopathogenesis of Canine Chronic Ulcerative Stomatitis.**
(DOCX)

**S4 Table. Complete data set.**
(XLSX)

## Acknowledgments

The authors are grateful for the technical expertise of the University of California, Davis School of Veterinary Medicine histopathology technicians. The primary author acknowledges the clinical expertise of Melissa Gates, DVM and the technical expertise of Crystal Holm and Danielle Stornetta, RVT.

## Author Contributions

**Conceptualization:** J. G. Anderson, B. G. Murphy.

**Data curation:** J. G. Anderson, A. Kol, B. P. Stapelton, K. Ford, D Vasilatis.

**Formal analysis:** J. G. Anderson, A. Kol, P. Bizikova, R. J. Jimenez, D Vasilatis, B. G. Murphy.

**Funding acquisition:** J. G. Anderson, B. G. Murphy.

**Investigation:** J. G. Anderson, B. G. Murphy.

**Methodology:** J. G. Anderson, P. Bizikova, A. Villarreal, B. G. Murphy.

**Project administration:** B. G. Murphy.

**Resources:** A. Villarreal, D Vasilatis, B. G. Murphy.

**Software:** R. J. Jimenez, D Vasilatis.

**Supervision:** B. G. Murphy.

**Validation:** A. Villarreal, B. G. Murphy.

**Writing – original draft:** J. G. Anderson, A. Kol, B. G. Murphy.

**Writing – review & editing:** J. G. Anderson, A. Kol, P. Bizikova, B. P. Stapelton, B. G. Murphy.

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
