## [Decision Letter · Decision Letter 0]

6 Nov 2019

PONE-D-19-21637

Immunopathogenesis of Canine Chronic Ulcerative Stomatitis

PLOS ONE

Dear Dr Anderson

Thank you for submitting your manuscript to PLOS ONE. After careful consideration, we feel that it has merit but does not fully meet PLOS ONE’s publication criteria as it currently stands. Therefore, we invite you to submit a revised version of the manuscript that addresses the points raised during the review process.

Many thanks for submitting your manuscript to PLOS One

After difficulties in getting appropriate reviewers, two reviewers were found

One recommended a rejection and the other recommended to accept

I have therefore decided on a major revision. If you feel that you can address reviewer one's comments then please do so and resubmit

I will send the resubmitted manuscript to the same reviewer.

I wish you good luck with your revisions

Thanks

Simon

We would appreciate receiving your revised manuscript by Dec 21 2019 11:59PM. To enhance the reproducibility of your results, we recommend that if applicable you deposit your laboratory protocols in protocols.io, where a protocol can be assigned its own identifier (DOI) such that it can be cited independently in the future. For instructions see: http://journals.plos.org/plosone/s/submission-guidelines#loc-laboratory-protocols

We look forward to receiving your revised manuscript.

Kind regards,

Simon Russell Clegg, PhD

Academic Editor

PLOS ONE

Journal Requirements:

2. Please include a copy of Table 1 which you refer to in your text on page 13.

Reviewers' comments:

Reviewer's Responses to Questions

**Comments to the Author**

1. Is the manuscript technically sound, and do the data support the conclusions?

Reviewer #1: Partly

Reviewer #2: Yes

2. Has the statistical analysis been performed appropriately and rigorously? 

Reviewer #1: Yes

Reviewer #2: Yes

3. Have the authors made all data underlying the findings in their manuscript fully available?

Reviewer #1: No

Reviewer #2: Yes

4. Is the manuscript presented in an intelligible fashion and written in standard English?

Reviewer #1: Yes

Reviewer #2: Yes

5. Review Comments to the Author

Reviewer #1: The authors sought to investigate some leukocyte populations by examining lesions from dogs with chronic ulcerative stomatitis. Nonetheless, there are many concerns described below:

1. The introduction is too long, and just in this section the authors cited 45 articles. I strongly suggest the authors to be concice.

2. Line 241. Why the authors used HE or toluidine blue? Why the authors did not standardize the dye used to stain mucosa tissue?

3. One of my major concerns are the antibodies used in immunohistochemistry. What is the specificify of the antibodies used for dogs? The cross-reactivity of all antibodies used must be proven (e.g. anti-CD20, anti-Mum1 and anti-CD204)? How the authors titrate the antibodies? All antibodies used to identify leukocyte populations are monoclonal antibodies (e.g. anti-CD20)? This information should be clearly stated. Why the authors did not include a negative control in immunohistochemistry assay? The controls of immunohistochemistry should be carefully conducted (please, for more details see: Hewitt et al., J Histochem Cytochem. 2014). How the authors select their antibodies? Why the authors used CD20 to identify B cells ("CD20 is expressed primarily on B cells but has also been detected on both normal and neoplastic T cells" available at https://www.thermofisher.com/order/catalog/product/RB-9013-P0#/RB-9013-P0? Why the authors used anti-CD204 (a macrophage scavenger receptor) antibody to identify macrophages?

4. Please, provide clearly the information about the monoclonal antibodies used to identify CD4+ and CD8+ cells. It is also a pity that CD4 and CD8 antibodies were not used together. Unfortunately, the importance of T CD4 CD8 double negative (e.g Fisher et al., Blood, 2005; D'Acquisto, Biochemical Phamarcology, 2011) and even positive (e.g. Clénet et al., Sci. Rep., 2019) cells was neglected in the present study. Furthermore, it would be great if the macrophage subsets has been considered.

5. The cross-reactivity of anti-human IL17 antibody must be determined/shown. The specifications of all antibodies used should be clearly stated (if possible, include the catalog number).

6. It is a pity that the authors did not quantify IL-17 by qPCR in tissue, and the producing cells is not clearly identified.

7. Line 389-390. The PloS One rules recommended that al data should be clearly showed.

8. The conclusion section should be profoundly revised.

Reviewer #2: This was a well designed and rigorously performed analysis of the pathological processes in this frustrating disease of the canine oral cavity. It is good to see mounting evidence of the likely immune-mediated pathway, in order that further research can hopefully lead to better methods of control or cure.

6. PLOS authors have the option to publish the peer review history of their article (what does this mean?). If published, this will include your full peer review and any attached files.

Reviewer #1: No

---

## [Author Response · Author response to Decision Letter 0]

6 Dec 2019

Response to reviewers has been uploaded. 

Table 1 is included in the manuscript.

Supporting information has been included at the end of the manuscript.

---

## [Decision Letter · Decision Letter 1]

18 Dec 2019

Immunopathogenesis of Canine Chronic Ulcerative Stomatitis

PONE-D-19-21637R1

Dear Dr. Anderson

We are pleased to inform you that your manuscript has been judged scientifically suitable for publication and will be formally accepted for publication once it complies with all outstanding technical requirements.

With kind regards,

Simon Russell Clegg, PhD

Academic Editor

PLOS ONE

Additional Editor Comments (optional):

Many thanks for your re submission to PLOS One

The manuscript was reviewed by the same reviewer as last time, and they have recommended acceptance of the manuscript

I wish to congratulate you on your research, and wish you all the best for the future

Many thanks

Simon

Reviewers' comments:

Reviewer's Responses to Questions

**Comments to the Author**

1. If the authors have adequately addressed your comments raised in a previous round of review and you feel that this manuscript is now acceptable for publication, you may indicate that here to bypass the “Comments to the Author” section, enter your conflict of interest statement in the “Confidential to Editor” section, and submit your "Accept" recommendation.

Reviewer #1: All comments have been addressed

2. Is the manuscript technically sound, and do the data support the conclusions?

Reviewer #1: Yes

3. Has the statistical analysis been performed appropriately and rigorously? 

Reviewer #1: Yes

4. Have the authors made all data underlying the findings in their manuscript fully available?

Reviewer #1: Yes

5. Is the manuscript presented in an intelligible fashion and written in standard English?

Reviewer #1: Yes

6. Review Comments to the Author

Reviewer #1: The authors performed a great job providing valuable information for the scientific community on the immunopathogenesis of canine chronic ulcerative stomatitis, and all my previous concerns were properly addressed.

7. PLOS authors have the option to publish the peer review history of their article (what does this mean?). If published, this will include your full peer review and any attached files.

Reviewer #1: Yes: Fernando Nogueira Souza

---

## [Editor Report · Acceptance letter]

27 Dec 2019

PONE-D-19-21637R1 

Immunopathogenesis of Canine Chronic Ulcerative Stomatitis 

Dear Dr. Anderson:

I am pleased to inform you that your manuscript has been deemed suitable for publication in PLOS ONE. Congratulations! Your manuscript is now with our production department. 

With kind regards,

on behalf of

Dr. Simon Russell Clegg 

Academic Editor

PLOS ONE